# A Longitudinal, Population-Level, Big-Data Study of *Helicobacter pylori*-Related Disease across Western Australia

**DOI:** 10.3390/jcm8111821

**Published:** 2019-11-01

**Authors:** Michael J. Wise, Binit Lamichhane, K. Mary Webberley

**Affiliations:** 1Department of Computer Science and Software Engineering, University of Western Australia, Perth 6009, Western Australia, Australia; 2School of Biomedical Sciences, University of Western Australia, Perth 6009, Western Australia, Australia

**Keywords:** *Helicobacter pylori*, gastric cancer, gastric disease, epidemiology

## Abstract

*Helicobacter pylori*, responsible for chronic ulcers and most stomach cancers, infects half of the world’s population. The Urea Breath Test (UBT) is one of the most accurate and reliable non-invasive methods for diagnosing active *H. pylori* infection. The objective was to use longitudinal, population-wide UBT data for Western Australia to look for *H. pylori*-related disease patterns. We collected 95,713 UBT results from 77,552 individuals for the years 2010–2015, likely representing all of the UBT samples analysed in Western Australia. Data collected also included sex, age and residential postcode. Other data reported here were inferred via a comparison with the 2011 Australian Census using a specially written Python program. While women appear to have more *H. pylori*-related disease than men, there is no difference in the disease rates once women’s higher rates of presentation for testing are taken into account. On the other hand, while the treatment strategy for *H. pylori* infection is generally very effective in Western Australia, failure of the first-line treatment is significantly more common in women than men. Migrants and Aboriginal Australians have elevated rates of *H. pylori*-related disease, while the rate for non-Aboriginal Australian-born West Australians is very low. However, no significant associations were found with other socio-economic indicators. We conclude that, for some people, *H. pylori*-related disease is not a solved problem.

## 1. Introduction

*Helicobacter pylori* is a gastric pathogen that affects half of the world’s population. *H. pylori* infection is acquired in childhood, though post-birth, with the onset of symptoms and complications usually occurring decades later [1,2,3,4]. The bacterium establishes infection by attaching to the stomach epithelium causing acute and chronic inflammation, described by Warren as active chronic gastritis [5]. The inflammation predisposes a patient to peptic ulcer disease and gastric cancer [6]. *H. pylori* infection rates vary across geographical regions. Poorer countries have higher rates of *H. pylori* infection and *H. pylori*-related disease is a major public health issue. However, it is now known that, even in wealthy countries, certain regions or ethnic groups can also present with high rates of *H. pylori*-related disease [7,8].

In Australia, *H. pylori* prevalence is relatively low, ranging from 15.4% to 30.6% [9,10], which is similar to most developed countries. By contrast, a previous study showed that Aboriginal Australians in Western Australia (WA) have *H. pylori* prevalence as high as 76% [11]. *H. pylori* infection prevalence has also been found to be high among migrants to Australia [12,13]. However, all the studies mentioned above were done on small samples at single time points. No longitudinal, population-wide study has been possible as *H. pylori* infection is not a notifiable disease.

A variety of tests are available for the diagnosis of *H. pylori* infection. The rapid urease test is highly accurate, but it is invasive, requires endoscopy, and is expensive [14]. On the other hand, the blood serum antigen test is inexpensive and non-invasive but cannot distinguish between past and current infections [15]. By contrast, the C${}^{14}$ Urea Breath Test (UBT) [16], which only requires a patient’s breath sample, is a simple diagnostic test and has a very high diagnostic accuracy [17]. Moreover, because of the extremely low quantity of ionizing radiation—well below background radiation—there is no restriction on the use of UBT, and it has been approved by US FDA for use by pregnant women and children [18,19]. It should be noted that there is a C${}^{13}$ Urea Breath Test [20] that has similar accuracy, but its use in Australia is limited, and effectively non-existent in Western Australia. That said, many of the findings of this study should apply equally to the C${}^{13}$ test as both C${}^{13}$-UBT and C${}^{14}$-UBT rely on the breakdown of urea produced by the patient via urease that is excreted by *H. pylori*. The resulting C${}^{13}$- or C${}^{14}$-labelled CO${}_{2}$ is then exhaled by the patient, and the amount can be measured.

The aim of the study was, through a series of statistical analyses based on large-scale UBT data, to better understand the epidemiology of *H. pylori*-related disease. This is the first time that longitudinal data for an entire population—the state of Western Australia—has been collected across a number of years.

## 2. Materials and Methods

### 2.1. Ethics

The study has been approved by the University of Western Australia (UWA) Human Research Ethics Office (RA/4/1/7952). Given the large number of samples from a variety of sources, and de-identified patients, informed consent was not required under the terms of the UWA Human Research Ethics Committee’s approval. However, the approval included a number of stipulations that have been adhered to.

### 2.2. Data

UBT data were kindly provided by the major pathology laboratories operating in Western Australia: The private laboratories Clinipath, Perth Pathology (now known as Australian Clinical Labs), St. John of God Pathology and Western Diagnostics, and the Western Australia Department of Health laboratory, PathWest. The data supplied by the laboratories, for each Urea Breath Test carried out over the years 2010 to 2015, was: A UBT value (an integer from 0 to approximately 12,000), the patient’s sex, year of birth and residential postcode, and the date of the test. Associated with each record was a stable identifier, provided by the pathology laboratory, so we can look for multiple tests involving the same patient. One laboratory provided names rather than identifiers, so a colleague (Dr. Chin Yen Tay) acted as data custodian, and created project-specific identifiers for that set of patients. Dr. Tay was not involved in the subsequent data analysis. To help ensure privacy, none of the identifiers used in this study had links to the system of Medicare identifiers used across Australia.

All the results reported below were inferred via statistical comparisons with data from the 2011 Australian Census provided by Australian Bureau of Statistics (ABS). The ABS provides data in the form of collections of statistical tables, called Data Packs [21]. A list of the tables used can be found in the Appendix A.

### 2.3. Methods

We assumed that all patients who received a UBT were experiencing dyspepsia or other gastric disease symptoms and were therefore being tested for *H. pylori* infection, in accordance with Maastricht recommendations [22]. Thus, from a detection of a *H. pylori* infection, we inferred the presence of *H. pylori*-related disease. Secondly, if there was more than one detection, the date of the first was used for later analyses. Starting with these assumptions, the analyses reported here were undertaken via a specially written computer program created in the Python programming language using the Python modules: Pandas [23] and scipy.stats [24].

The first task was to clean the data, so records relating to tests for people living outside of Western Australia were deleted because we lacked substantial corresponding data for other states. Records without numerical UBT values were also deleted.

The next step was to map patients living at residential postcodes (i.e., not postcodes for post office boxes) to the Level 2 Statistical Areas (SA2) used by the Australian Bureau of Statistics in their Data Packs. The ABS maps regions of Australia across a range of granularities, ranging from very small regions containing on average 400 people (SA1) to very large regions containing more than 100,000 people (more in metropolitan areas), SA4 [25]. For most analyses in this study, we were only interested in SA2 areas with at least 500 residents. Postcodes can overlap multiple SA2 areas, so each unique patient was coded as 1.0 UBTp (UBT person), so if the patient’s residential postcode corresponded to more than one SA2, the UBTp was allocated pro rata to the multiple SA2 areas based on relative population sizes. That is, if the two SA2 areas corresponding to a postcode have 20,000 and 30,000 people, the 1.0 UBTp was allocated 0.4 and 0.6, respectively, to the two SA2 areas. All the UBTp across the 6 years was treated as a single pool.

The method used in most of the analyses was to compute, across eligible SA2 areas (i.e., minimum population 500), the sum of attributed UBTp based on the populations of the particular group across respective SA2 areas, and compare these with the expected proportion of UBTp from the state as a whole using a binomial distribution statistic. For example, in the analysis of positive UBT findings in Aboriginal Australians compared to non-Aboriginal people, the sum was computed across all the SA2 areas of the percentage of the UBTp attributable to Aboriginal Australians, based on data from the Census. For example, in the SA2 area 51,085, there were 18,044 people at the 2011 Census, of whom 738 identified as Aboriginal Australians. The UBTp value for this SA2 area was 180.389, so the proportion attributable to Aboriginal Australians is 7.378. The sum of UBTp attributable to Aboriginal Australians across all the SA2 areas was then compared, using a binomial distribution statistic, to the expected value, which is the percentage of the total UBTp pool attributable to Aboriginal Australians as a percentage of the Western Australian population.

### 2.4. Data Availability

The underlying data, and the program used to analyse that data, can be found at https://doi.org/10.26182/5db91300c6031.

## 3. Results

### 3.1. Overview

A total of 95,713 UBT results from 77,552 individuals were collected and analysed. In addition, 15,903 individuals (20.5%) were found to be UBT positive, where, for the C${}^{14}$ test, a UBT count of 200 or more is regarded as UBT positive, and a score of 50 or less is regarded as UBT negative. If there was more than one detection (UBT ≥ 200), dating was based on the date of the first detection. A UBT score between 51 and 199 is regarded as borderline, which implies that the test needs to be redone [26]. In addition, 7.4% of the first Urea Breath Tests were borderline, but only 40.00% of people with an initial borderline result had a follow-up test recorded.

Out of the 15,903 UBT positive individuals, 15,760 had postcodes that could be be mapped to Level 2 Australian Bureau of Statistical Areas (SA2). Based on the WA population, the incidence of disease related to *H. pylori* infection was 645.76 per 100,000 individuals over the six year period (2010 to 2015), or 107.63 per 100,00 individuals per year. Viewed by SA2 areas, the incidence of disease related to *H. pylori* infection ranged from 424.1 UBTp (Dianella), 303.3 (Bayswater) and 263.9 (Bentley), down to 3.7 (Gidgegannup, 3.6 (Wattleup) and 3.3 (North Coogee), over the six-year period. The complete list of SA2 areas with more than 500 people, and their counts of UBTp, can be found in the Appendix A.

It is worth noting that, for most of the analyses in this study, we have assumed that both the population of Western Australia and the count of initial *H. pylori* infection detections per year were largely static over the period of interest, fixed at the level of the 2011 Census. This is not strictly true. While we only have comprehensive data for the 2011 Census, the Australian Bureau of Statistics publishes annual population estimates, from which we determined via a linear regression that the population of Western Australia grew by 2.18% per year over this period. In addition, the median age dropped very slightly, from 33.188 to 33.113, and for the first four years of the period, the dominant source of increase was migration from overseas. The population of Western Australia in 2015 was 2,590,259. Interestingly, over the period, the incidence of *H. pylori*-related disease rose by 14.55% per year, from 2022 in 2010 to 3315 in 2015.

### 3.2. Sex and UBT

Six UBT positive individuals had no sex recorded (and 27 out of the starting count of 77,552 individuals). These records were not used in the computations involving sex. Of the 15,897 UBT positive individuals for whom sex was recorded, 9216 were women, versus an expected count of 7901.7. Based on women as a percentage of the Western Australian population, this is highly statistically significant (*p*-value: $2.83\times 10^{-97}$, on a binomial distribution statistic). However, this computation does not exclude the possibility that more women came to be tested, and therefore more infections were found. To examine this possibility, we defined presentation for UBT testing as the percentage of patients—presumably with gastric symptoms—whose first (and generally only) test came back as negative. The percentage of women presenting for UBT testing was 59.06%, so, while women appear to be significantly over-represented among all those with positive UBT determinations (57.8%), once women’s higher presentation for UBT testing is taken into consideration women were not significantly more likely to be infected than men (*p*-value > 0.05). In addition, using a binomial distribution statistic, we found a small, but statistically significant difference between men and women in the mean age when infection is first detected: 44.58 years in men versus 46.05 years in women (*p*-value: $3.736\times 10^{-7}$).

### 3.3. Efficacy of Treatment and Gender Bias in Resistance

We looked at the individuals with two or more positive UBT determinations, as this suggests treatment failure, and assumed that the second test will have followed treatment. Only 2719 people (64.9% women) had more than one positive UBT determination, which implies that the first-line, triple therapy appears to work most of the time (82.90%). However, even when women’s higher rate of presentation for testing was taken into account, women are significantly over-represented (*p*-value: $2.17\times 10^{-10}$, on a binomial distribution statistic). Counts of positive tests per person, rather than counts of persons with at least two positive tests, yielded a similar statistic.

We then drilled down to look at the profiles of those with multiple positive tests. Both men and women had a median of two tests (respective means 2.5675 and 2.5493), which implies that, if the initial treatment fails, the efficacy of subsequent treatments (often quadruple therapy) is not significantly different in women versus men. In other words, 93.88% of patients were cured of *H. pylori* infection by the first line or follow-up treatments. However, for some patients, it could take much longer. The maximum number of positive tests for men was 9, implying eight failed treatments; the corresponding maximum number of tests for women was 18.

### 3.4. H. Pylori Related Disease in Aboriginal Australians

We compared the number of Aboriginal Australian UBTp across the SA2 areas, with the number predicted as a percentage of Western Australians from the 2011 Census. The number of Aboriginal Australians in Western Australia at the 2011 Census was 69,185 (3.1% of the population of Western Australia). Taken across SA2 areas with population greater than 500, 548.183 UBTp were attributable to Aboriginal Australians, but, based on Aboriginal Australians as a percentage of the Western Australian population, 488.687 UBTp would be expected, which is a significant over-representation, with a *p*-value of 0.00343 on a binomial distribution statistic. However, when we once again used the presentation for testing metric—this time for Aboriginal Australians rather than women, as a percentage of the WA population—the presentation for testing percentage for Aboriginal Australians reduced to 2.8%, which means that the *p*-value is now even more significant, $4.99\times 10^{-8}$.

### 3.5. H. Pylori Related Disease in Non-Aboriginal Australians and People Born Overseas

The 2011 Australian Census provided country of birth data for Australia and 33 countries, together with the “Born Elsewhere” and “Country not Stated” categories. Given that we had already learnt that there is a significant level of *H. pylori* related disease in Aboriginal Australians, we wanted to compare the rate for non-Aboriginal people born in Australians compared with people born overseas. After first determining that the people who did not declare a country of birth appeared to be proportionally split between people born in Australia and people born overseas, UBTp attributable to “Country not Stated” was split proportionally, and the Aboriginal Australian component of was deducted from the Australian born UBTp. From this analysis, *H. pylori*-related disease is significantly over-represented in people born overseas (expected UBTp 5145.00, attributed UBTp 5544.63, *p*-value: $4.83\times 10^{-12}$), while it is significantly under-represented in non-Aboriginal Australian born people (expected UBTp 10,064.84, attributed UBTp 9835.15, *p*-value: $4.64\times 10^{-5}$), all based on a binomial distribution statistic.

### 3.6. Socio-Economic Indicators and H. pylori-Related Disease

From the 2011 Census, we were able to get data on the highest level of schooling: Year 8, Year 9, Year 10, Year 11, and Year 12; people who did not attend school; people with technical qualifications; people with university qualifications, and people who were out of work on the day of the Census. Only two of these possible explanations had statistically significant associations at the 0.05 level with *H. pylori*-related disease, and then only weakly. People who did had not attended school were associated with increased *H. pylori*-related disease (*p*-value 0.0287, based on small numbers: 75.6 UBTp expected versus 92.7 UBTp inferred), while gaining university level qualifications was associated with reduced incidence of *H. pylori*-related disease (*p*-value 0.0164). None of the remaining indicators showed a significant association at the 0.05 level.

### 3.7. Age and UBT

Two UBT positive individuals only had a default Date of Birth (1900) recorded, making them at least 110 years old, so were excluded from the age related calculations. When the UBT positive individuals were placed in age ranges to allow comparison with the census data for Western Australia, we see that the disease incidence is significantly associated with people from age groups spanning 25 to 84 years (Figure 1 and Appendix A). This implies that *H. pylori*-related disease is not mainly found in older people, as has previously been believed. On the other hand, *H. pylori* related disease is largely absent in the young.

We then defined the borderline ratio as the number of borderline results divided by the number of determinate (i.e., positive plus negative) results, here reported for the different age ranges. Patients in age groups 5–14 years and 85 years and older were found to have a much higher borderline ratio than the middle age ranges (Appendix A). Moreover, we noticed that, for the very young and very old categories, the median negative value was less negative (closer to boundary count of 50) while the median positive was less positive, though still well above the threshold for positive determination (200) (Appendix A).

## 4. Discussion

The Urea Breath Test was invented 30 years ago and is still one of the best diagnostic methods for *H. pylori* infection because of its high diagnostic accuracy, non-invasiveness [17,26] and ability to retest following treatment, unlike, for example, tests based on serology. We used population-wide UBT data for the state of Western Australia to estimate the prevalence of disease related to *H. pylori* infection across the state. We believe that the UBT data represent all the UBT conducted in Western Australia over those years, as the collection includes data from all the diagnostic laboratories operating in WA.

### 4.1. Limitations

We acknowledge several weaknesses in our study. Each patient only had one or more UBT results; the ethics approvals underlying the methodology preclude a reference method to prove that a positive test result represents a genuine infection, or that a negative result genuinely represents freedom from infection. (That said, UBT has a sensitivity of 0.96 and specificity of 0.91 [17]). Secondly, we have assumed that each presentation for testing was due to the presence of gastric symptoms, for which *H. pylori* infection could be a possible explanation. However, it is also possible that a person who had been diagnosed with the infection may arrange for his/her asymptomatic family members to also be tested. Unfortunately, from the anonymised data we have been given, there is no way to estimate the extent of the testing of otherwise asymptomatic individuals. Similarly, we cannot discern whether patients are attending multiple doctors, where the change of doctor has resulted in a change of pathology service. (Each pathology laboratory used their own system of patient identifiers.)

### 4.2. Prevalence and Demographic Results

Based on the UBT data, the prevalence of *H. pylori*-related disease was found to be 22% of those experiencing gastric symptoms. This is similar to the levels of *H. pylori* carriage that have been reported for developed countries, e.g., 24.6% for Australia [27], but it needs to be born in mind that our UBT data were collected from people experiencing gastric symptoms rather than from a screening study of randomly selected volunteers.

Our finding of a significant presence of *H. pylori*-related disease among Aboriginal Australians, and under-representation of the disease in non-Aboriginal Australians, mirrors previous work on a much smaller sample, which found a high level of *H. pylori* infection in Aboriginal Australians (76%) compared with the non-Aboriginal Australian populations (30%) [11]. Similarly, we found a high rate of *H. pylori*-related disease among migrants to Australia, but a significantly lower rate for non-Aboriginal people born in Australia, which mirrors a similar result by a team in Sweden that tested for infections in 36 classes of primary school students, and observed an *H. pylori* infection rate in children with Swedish parents of 2%, but 55% in children originating from the Middle East and Africa [28].

In contrast, it is interesting that, unlike earlier studies which pointed to a range of socio-economic factors, e.g., income levels, smoking [29] and low education level [30], we found no significant association with any of the socio-economic factors for which we had data.

Recall that *H. pylori* infection is usually acquired in childhood, with the onset of symptoms and complications usually occurring much later in life. Our data in Figure 1 and Appendix A show that the onset of *H. pylori* related disease occurs earlier than generally believed, with significant over-representation of people in the 25–34 age range among those with positive UBT results versus their percentage of the the WA population. Our data also show an unexpected under-representation of positive determinations in the elderly (85 years and older).

### 4.3. Sex Bias in H. pylori Disease

A number of studies have examined whether there is a sex-bias in the incidence of *H. pylori* infection. Many have found a higher rate in men, e.g., [29,31], and a large study in China of people undergoing health checks [32]. On the other hand, the EUROGAST project undertook a random sampling study of 3194 people from 17 populations in Europe, North Africa, North America, and Japan found no significant difference (though there was a slight bias to male). It is therefore interesting that the data from Western Australia had an apparent bias to *H. pylori*-related disease in women, which disappeared once the rate of presenting for testing was taken into account. However, our study also found that male patients present with symptoms 18 months earlier than female patients. Given that women are more likely to come forward for testing, this could also imply that the symptoms are more pronounced in men. While the earlier onset in men needs to be investigated further, some support for this comes from a study which compared results from a community survey of gastroesophageal symptoms, including reflux, with patients presenting for anti-reflux surgery. While that study was primarily focused on outcomes post surgery, it is worth noting that a lower proportion of men in the community survey reported never having symptoms of heartburn, though the proportion of women was higher than men in the most severe categories (daily, and multiple times per day). In addition, the mean age of men presenting for surgery was also markedly less than women—48.0 years for men compared with 55.4 years for women [33].

### 4.4. Efficacy of H. pylori Treatment and Bias in Treatment Failure

In analysing the patients who had multiple positive UBT results, we assumed that the patients had undergone standard antibiotic treatment between each UBT. Based on this assumption, the data suggest that the first line triple therapy, which consists of a proton pump inhibitor, amoxicillin, and clarithromycin [34], remains effective in Western Australia, with 82.9% of patients not having a second positive UBT result. Although the treatment failure is low, our study suggests that treatment fails more often for women than men. A similar finding was reported in a study by Moayyedi et al. [35], though there was no evidence in that study as to why this was occurring.

Should the first line treatment (presumably the clarithromycin and/or metronidazole based triple therapy) fail, the subsequent salvage therapy is usually a bismuth-based quadruple therapy that includes a Proton Pump Inhibitor, bismuth, and two antibiotics [22]. Since there is no firm guideline regarding which antibiotics to use after the failure of the first-line triple therapy, the choice of the two replacement antibiotics, depending on antibiotic resistance profile reported in the local region, can be either tetracycline, rifabutin, furazolidone, or levofloxacin [22]. Despite the first line therapy failing more often in women than in men, the efficacy of the subsequent treatment was not found to be different between the sexes (mean 2.55, median 2 positive tests for women, and mean 2.57, and median 2 positive tests for men).

## 5. Conclusions

In this study, we have taken a big-data approach to examine Urea Breath Test data from across Western Australia for the years 2010–2015. The existence of a stable (anonymised) identifier for each patient has meant that we were able to look at the data both as a single snapshot, but also longitudinally. We can take a number of messages from the approach used and the findings:*H. pylori* infection is largely seen as a solved problem. However, we were able to find groups within the community who are disproportionately affected by *H. pylori*-related disease, notably Aboriginal Australians and migrants to Australia. There are also some individuals in whom the infection is particularly refractory.While the number of people with borderline test results is small, the majority of people with an initial borderline test result did not have a follow-up UBT. As some of these are likely to have been positive, there is cause for concern.In both personal and health-economic terms [36], it is better to test and treat the infection, and thereby eliminate the disease [22]. This remains the case even for early stage gastric cancer [37]. The approach we have taken highlighted infection hotspots (seen in Appendix A). Similar analyses could help planners target education about test-and-treat to physicians and their patients.The open questions thrown up by this study are: Why is it that the first-line treatment fails significantly more often for *H. pylori* infection in women than in men, but why are men presenting with symptoms significantly earlier than women; are their symptoms worse?

## Figures and Tables

**Figure 1 jcm-08-01821-f001:**
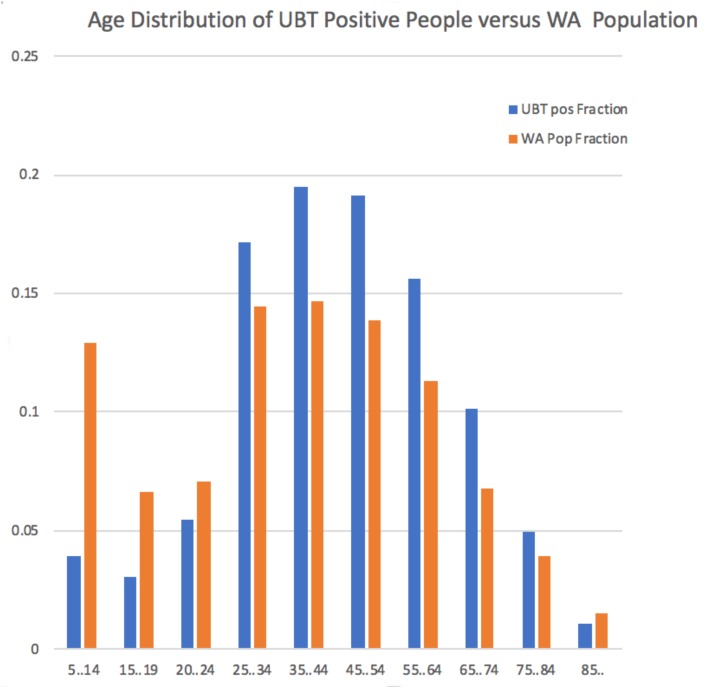
*H. pylori* positive results by age group. Percentage of UBT positive individuals across age ranges spanning 5 years to 85 years and older, compared with the percentage of individuals for the respective age ranges based on the 2011 Australian Census data for Western Australia. The age ranges are those used by the Australian Bureau of Statistics in reporting results from the Census.

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
