# Peer review of "A Longitudinal, Population-Level, Big-Data Study of *Helicobacter pylori*-Related Disease across Western Australia"

_jcm, 2019, doi:10.3390/jcm8111821_

Round 1
Reviewer 1 Report
Dear authors and editor,
The manuscript titled ‘’ A longitudinal, population-level, big-data study of Helicobacter pylori-related disease across Western Australia ‘’ is the first t longitudinal data for an entire population – the state of Western Australia which tray to better understand the epidemiology of H. pylori related disease. The results are interesting and confirm previous studies on a small group of patients from Australia.
The paper is well-organised, the language is correct and the content is understandable. Literature properly selected and mostly up to date. I believe they add some contribution to the literature.
However the manuscript is good, I have some comments that should be clarified.
The Python program that implements the analyses is incomprehensible for me(I’m surgon). In my opinion, in 5% of cases, the results should be confirmed by a test other than UBT. The UBT test is good although it also has disadvantages. It is more suitable for control after the implementation of HP eradication. The result (20.5%- infected ) seems to be underestimated. The discussion part requires extension and refinement .
The lack of these elements definitely reduces the value of work.
In conclusion, the manuscript needs some improvements.
Thank you for your choice me as a reviewer.
Author Response
While Reviewer 1's generally positive response to the manuscript is appreciated, I think it needs to be born in mind that the evidence here is purely from epidemiological inferences based on a very limited set of data per patient: Urea Breath Test result (a number), sex, date of birth, date of the test and patient's residential postcode. On the other hand, the data set is the largest we know of, for a whole population (not a sample) and spanning several years.
From this, data about infections in Aboriginal Australia, for example, were inferred via a comparison with 2011 Australian Census data. There was no contact with patients, so no way of knowing what actually happened; all we can do is infer what happened. For example, from the fact that someone comes to be tested we infer the presence of symptoms. From more than one positive result we infer that there was treatment after the first test, but that the treatment failed. Otherwise, why have a second test? In other words, it well beyond our data (and ethics clearance) to find out whether a positive test did indeed reflect a real infection. Even more problematic is to test someone who is UBT negative to discover whether they really were negative, ie false negative. All we can do is refer you to the accuracy and precision results quoted in the literature for the UBT. You may find it interesting to look at the Maastricht/Florence V guidelines (Malfertheiner et al, Gut, 66:6-30)
Reviewer 2 Report
This is an interesting study of the occurrence of Urea Breath Test (UBT)- detected H.pylori in Western Australia. It requires major re-formatting to bring it to acceptable format.
P1 L39-30 Helicobacter pylori laboratory. Please explain why relevant, or delete
L30: pls provide prevalence for non-Aboriginal Australians
L30 ... immigrants.. (where? in Australia? pls be specific)
The aims of the study L47-48 intro, are not informative. Please spell out the aims at the end of the introduction.
P2 L50 -51. ...test and treat.. people with dyspepsia.
L51 Treatment eliminates HP infection (not 'the disease')
Methods. Please explain all the definitions of all the exposures studies. What demographic and lifestyle data did the authros have per patient. This infromation is 'sprinkled throughout the paper. A lot of effort is spent on somewhat trite explanations of pro-rata calculations (e.g. L81+that whole section can be condensed in 2 sentences). Statistical methods should be elaborated on. Why that method? Were 95% confidence limits or tests of significance included, etc.
Define what you are measuring. Is UBT measuring incidence or prevalence? the discussion seems to swop from one to the other.
Stick to one definition of who gets tested. is it people with dyspepsia or gastric symptoms.
L69 explain why synthetic identifiers were used and how
L71 'All other data' Please specify
L72 What is a 'data pack' - why is it relevant?
Results. This is a mixture of results and discussion, bordering on a data-fishing exercise.
L81-83. Everyone cleans their data - was there anything relevant worthy of mention? What is the proportion of missing data. How were missing values dealt with in the analysis? Was there any sensitivity analyses done by including some of the missing data?
Please explain what 'panda' and 'scipy.stats' do.
L85 Please define what an SA2 area is for an international audience.
What is the proportion of data deleted. How many people had postboxes?
L115. ... per 100,000
Results. Please provide a graph / table of the prevalence of UBT positive results (in people with dyspepsia) by age group, and the other demographic characteristics studied.
Discussion requires shortening, and focus.
Author Response
> P1 L39-30 Helicobacter pylori laboratory. Please explain why relevant, or delete
Done
> L30: pls provide prevalence for non-Aboriginal Australians
The only data that has been published till now has been for Australia as a whole, or for Aboriginal Australians.
> L30 ... immigrants.. (where? in Australia? pls be specific)
Changed to migrants to Australia, in line with the associated references
> The aims of the study L47-48 intro, are not informative. Please spell out the aims at the end of the introduction.
Done. The text was simplified and clarified.
> P2 L50 -51. ...test and treat.. people with dyspepsia.
“uninvestigated dyspepsia” is a phrase from the Maastricht V/Florence guidelines (to indicate a situation where there are symptoms, but no investigation or treatment has been attempted thus far). The phrase no longer appears as such in the manuscript, but in any case, the related discussion was moved to the Discussion section
> L51 Treatment eliminates HP infection (not 'the disease')
Changed the phrasing so that it is the infection that is being eliminated, and thereby the disease.
> Methods. Please explain all the definitions of all the exposures studies.
All the analyses used the same method, which is outlined in the Methods section: Add up the UBTp for each SA2 with at least 500 people, and then compare with expected values using a binomial distribution statistic. It would be boring in the extreme to say essentially the same thing for each analysis, which is what I assume the Reviewer was referring to.
> What demographic and lifestyle data did the authros have per patient. This infromation is 'sprinkled throughout the paper.
Actually, none. The only data we had access to is described in the Data subsection: “a UBT value (an integer from 0 to approximately 12,000), the patient's gender, year of birth and residential postcode, and the date of the test. Associated with each record was a stable identifier, provided by the pathology laboratory, so we can look for multiple tests involving the same patient.” All the other data was inferred by the program, as described in the other sections
> A lot of effort is spent on somewhat trite explanations of pro-rata calculations (e.g. L81+that whole section can be condensed in 2 sentences). Statistical methods should be elaborated on. Why that method? Were 95% confidence limits or tests of significance included, etc.
The point of that section was to explain the method that was used, using the calculation involving Aboriginal Australians as the example. I have also gone through and emphasised that a binomial distribution statistic is being used, which does not lend itself to confidence intervals, compared with odds ratios, say.
> Define what you are measuring. Is UBT measuring incidence or prevalence? the discussion seems to swop from one to the other.
Fair point. Mostly incidence, but in a few places it really is prevalence. I have now checked, and if necessary, revised each such usage.
> Stick to one definition of who gets tested. is it people with dyspepsia or gastric symptoms.
The fact is that we have no idea. The point being made is that there must have been some reason why a UBT was sought, hence the inference of symptoms. This is reworded as dyspepsia or other gastric symptoms
> L69 explain why synthetic identifiers were used and how
There is now text to explain why project specific identifiers were used (related to ethics of patient confidentiality, so the IDs cannot be traced back to individuals, as the Medicare number can.
> L71 'All other data' Please specify
I have reworded that to “All the results reported below”. (Data was being used in the sense of output, computed data rather than input data.)
> L72 What is a 'data pack' - why is it relevant?
That is now explained.
> Results. This is a mixture of results and discussion, bordering on a data-fishing exercise.
I guess “data-fishing” is in the eye of the beholder. Each analysis was an experiment that looked at different statistical attributes that could be computed from the data, mostly based on the literature, e.g. by sex, or migrant to Australia version Australian born. The results describe what was found. The discussion puts these findings into context – quite traditional.
> L81-83. Everyone cleans their data - was there anything relevant worthy of mention?
Yes. For transparency. Of course, all of this is in the program, which is available for download.
> What is the proportion of missing data.
Essentially none. Records with missing values were not used in analyses involving those values. For example, if sex was not stated, the record could not be used in comparisons of women versus men, but were used for computation of rates for Aboriginal versus non Aboriginal Australians, say.
> How were missing values dealt with in the analysis? Was there any sensitivity analyses done by including some of the missing data?
For each analysis, e.g. age, sex, place of birth, there is now a statement about how many data records were ignored. (In the original manuscript, this was only done for the place of birth data, mapping from postcodes). The numbers are tiny.
> Please explain what 'panda' and 'scipy.stats' do.
These are Python modules. That is now made explicit, with a URL to each
> L85 Please define what an SA2 area is for an international audience.
Text describing Australian Bureau of Statistics Statistical Areas has now been added
> What is the proportion of data deleted. How many people had postboxes?
143 (15903 UBT positive minus those that could be mapped, 15,760)
Done
> L115. ... per 100,000
Fixed
> Results. Please provide a graph / table of the prevalence of UBT positive results (in people with dyspepsia) by age group, and the other demographic characteristics studied.
Figure 1 and a table in the Supplementary Data show the incidence across age ranges. The other data took the form of binary, comparisons, e.g. inferred incidence in Aboriginal Australians versus non-Aboriginal Australians, men versus women. These do not lend themselves to graphics, or even really tables.
> Discussion requires shortening, and focus.
I'm not sure how to do that without losing some aspect of the findings, which is the point of the paper. Each section of the results records the output from a single experiment undertaken in the Python code. The corresponding Discussion section puts those results into context of the literature. Other than being computational, this is really no different to a more traditional bioscience paper which discusses, say, sequenceing of gene, cloning into a vector, test for DNA and protein expression, etc.
Overall, I have gone very carefully through the paper and sought to focus the telling of the story as it's as concise and user-friendly as possible.
Round 2
Reviewer 1 Report
Dear authors and editor,
The manuscript titled ‘’ A longitudinal, population-level, big-data study of Helicobacter pylori-related disease across Western Australia ‘’ is well-organised, the language is correct and the content is understandable. Literature properly selected and mostly up to date. I believe they add some contribution to the literature.
Authors' answers to the questions in the review do not solve all my questions. However, I understand that they worked only on statistical data and this is the weak point of the project. In my opinion, each result (method) should be confirmed by another method in a few percent of causes. Then it is 100% reliable. I do not notice changes in the revised manuscript (should be highlighted in a different colour).
Thank you for your choice me as a reviewer.
Author Response
In lab based biological studies it is entirely routine for a study to use a method, often supplied in kit form, with a comment appearing in the paper similar to: "The method was applied as per manufacturer's instructions ", (followed by the name of the manufacturer or a citation to the literature). Unless it is the point of the research, one does not reevaluate the method, but rather uses it to reveal some new science.
In this study we have taken as read the quoted accuracy and specificity of the C14 Urea Breath Test, and then statistically mined large-scale data derived from use of that method. It was not the purpose of this study to reevaluate the method. Indeed, the experimental design precludes that option because it would be unethical to conduct an invasive confirmatory biopsy on the 80% of patients with an initial negative finding, but how else can you test for false negatives? Of those with initial positive UBT results, we infer that ~80% are successfully treated, so again are unlikely to have a confirmatory biopsy. In general, it is likely that only for the most refractory infections that a biopsy will be undertaken, but by now the data set will be skewed to only true positives. In short, while it may be useful to reexamine the accuracy and specificity of the UBT method, any study based purely on clinical data will be unable to estimate the rate of false negatives, and may be hampered in examining the rate of false positives by bias in the set of positive findings that go on to be further tested.